# Running in Natural Spaces: Gender Analysis of Its Relationship with Emotional Intelligence, Psychological Well-Being, and Physical Activity

**DOI:** 10.3390/ijerph19106019

**Published:** 2022-05-15

**Authors:** Yolanda Campos-Uscanga, Hannia Reyes-Rincón, Eduardo Pineda, Santiago Gibert-Isern, Saraí Ramirez-Colina, Vianey Argüelles-Nava

**Affiliations:** 1Instituto de Salud Pública, Universidad Veracruzana, Xalapa 91190, Mexico; ycampos@uv.mx (Y.C.-U.); han_nia29@hotmail.com (H.R.-R.); 2Red de Biología y Conservación de Vertebrados, Instituto de Ecología A.C., Xalapa 91070, Mexico; eduardo.pineda@inecol.mx; 3Dimensión Natural S.C., Coatepec 91608, Mexico; santiagogibert75@gmail.com; 4Sistema de Atención Integral a la Salud, Universidad Veracruzana, Xalapa 91020, Mexico; sarramirez@uv.mx

**Keywords:** health-promoting environments, green exercise, running, emotional intelligence, psychological well-being, body dissatisfaction

## Abstract

Running is a complete and accessible physical exercise for the population, but little research has been done on the psychological and environmental variables related to its practice. The objective of this research was to determine how emotional intelligence, psychological well-being, and body dissatisfaction are related to running in natural spaces for men and women. A cross-sectional study was conducted on 331 runners from 20 states of the Mexican Republic (55.3% women), between 18 and 80 years old (m = 37.4; SD = 11.5), with an average of 7 years running experience (SD = 9.3). The Brief Emotional Intelligence Inventory, the Psychological Well-Being Scale, and the Body Shape Questionnaire were used. The results show that men who run in natural spaces have greater psychological well-being and emotional intelligence (stress management) and less body dissatisfaction, and they run more days per week than those who run in built spaces. Predictors of running in natural spaces were greater psychological well-being and emotional intelligence (stress management). On the other hand, women who run in natural spaces show lower emotional intelligence (stress management) and run for more minutes per day. The predictors for running in natural spaces were identified as lower emotional intelligence (stress management), running for more minutes per day, and practicing another physical exercise. In conclusion, in this heterogeneous sample, natural environments are likely to be related to better performance and certain psychological indicators for runners. However, these relationships differ between men and women, so further studies with larger sample sizes are needed to confirm our findings.

## 1. Introduction

One of the main priorities in public health today is finding strategies to promote physical activity, especially because of the evidence that its practice improves people’s health. Physical exercise contributes to better health, disease prevention, and control of chronic non-communicable diseases such as diabetes, hypertension, heart disease, and obesity [1]. At the same time, it contributes to mental health by reducing anxiety, depression, and stress and improving cognitive attributes, such as social skills, psychological well-being, self-concept, self-esteem, and the ability to overcome traumatic circumstances [2]. Therefore, scientific evidence is needed to understand how physical exercise can help in achieving better population health outcomes.

Living on a street with vegetation is associated with a higher likelihood of using active transportation (e.g., cycling) and walking; distance from green spaces is inversely associated with the likelihood of physical activity [3]. Additionally, those who perform physical activity in natural spaces report greater enjoyment [4], so one can assume that the environment is an extrinsic motivator for physical activity, being key both to starting and continuing with its practice [5]. Similarly, exposure to urban green spaces has been found to have a negative association with mortality, heart rate, and violence, and a positive association with physical activity, attention, and mood [6].

The evidence available today shows that the benefits of physical activity are not limited to humans, as green exercise can also lead to the protection and sustainability of natural environments and promote planetary health. Green exercise makes it possible to increase the population’s physical activity, while also reducing the environmental impact of human activities [7]. Increased physical activity in natural spaces, together with educational processes leading to respect and care for the environment, result in a rise in the value that people assign to these spaces and, consequently, in an increase in the care for the species that inhabit such spaces. Additionally, they encourage active mobility and reduce the number of motor vehicles used for transportation, thus reducing environmental pollution and traffic accidents, as well as improving efforts to mitigate climate change [8].

When physical activity involves exposure to natural environments, it provides special contributions for improving mental health [9]. Interaction with nature has also been shown to be related to greater psychological well-being [10], especially when mediated by physical activity [11]; however, although it is known that greater psychological well-being is related to greater athletic performance [12], the influence of the environment in this regard has not been investigated. There is evidence, however, that the influence of natural spaces is not generalizable to all types of physical activity. For example, a study comparing golfers and walkers in natural environments found that for walkers, the natural environment stimulates practice and generates positive feelings, while golfers, on the other hand, perceive natural elements as obstacles to better performance [13]. Therefore, it is essential to analyze the influence of the environment for each type of physical activity.

It has also been found that for adults raised in areas of lower economic incomes, emotional intelligence is positively related to the greenness of the areas of residence. However, this relationship is negative in adults raised in areas of higher economic incomes [14]. On the other hand, it has been found that people with higher emotional intelligence have greater psychological well-being [15] and engage in more leisure-time physical activities [16]. Available evidence shows that emotional intelligence may play an important role in individuals’ physical activity patterns [17] and even in improving sports skills [18].

However, the aforementioned relationships may have differences when analyzed by sex. Given that, in terms of body composition, there are differences in the proportion of fat mass and muscle mass between the sexes [19], it is expected that there will be differences in exercise performance, which may be accentuated by gender aspects and traditional roles. It has been observed that more men engage in physical exercise than women, and men also dedicate more time to the practice than women [20]. Women present greater body dissatisfaction [21] and report lower scores of psychological well-being in the dimensions of self-acceptance and autonomy; however, in positive relationships with others and personal growth, they score higher than men [22], while men obtain higher scores in emotional intelligence [16,23].

Furthermore, women make more frequent use of parks and natural areas [24]. The green space health associations were stronger for women than for men in Europe and North America, but not in other continents [25]. Therefore, it is necessary to further study the relationship between green spaces and health outcomes in populations, such as in the Mexican population.

Considering that physical exercise may require implements, equipment, or spaces specially designed for its realization, some practices are of limited access to certain population groups. Running is a type of physical exercise that is complete and accessible to the general population, since it requires fewer implements than others and many spaces are available for its realization. Despite this, physical inactivity rates in Mexico are highest in the population aged 18 years and older: 66.7% of women and 53.3% of men reported not engaging in any sport or physical exercise [26]. Therefore, some questions arise about the characteristics of athletes and the environments in which they exercise.

Research indicates that the main motivations that runners identify for running are social and psychological interactions, in addition to improving their health and physical conditions [27]. In turn, psychological well-being is positively related to satisfaction with body image [28]. The above allows hypothesizing about the potential of running in natural spaces to contribute to improved psychological well-being and body image satisfaction.

In the context of the SARS-CoV-2 pandemic, research on nighttime runners in Poland found that 31.5% reported that running reduces their stress levels, but only 10.5% prefer to run in natural spaces [29]. This may be explained by the time of day during which they run, as darkness may increase the risks involved in running in natural areas. A study of Mexican runners revealed that men showed better running performance than women. In addition, those who ran frequently in natural spaces showed less body dissatisfaction and greater psychological well-being than those who ran infrequently in such spaces. A negative association was also found between psychological well-being and body dissatisfaction [30]. Although this study may provide some clues about what happens with runners, the associations were not made with distinctions according to sex and the space in which they exercised.

A deeper understanding of the emotional and cognitive mechanisms underlying running is required, with separate analyses by sex. Therefore, the present study aimed to determine how emotional intelligence, psychological well-being, and satisfaction with body image in men and women are related to running in natural spaces. The hypothesis was that, in both men and women, runners who run in natural spaces present greater emotional intelligence, psychological well-being, and satisfaction with body image than those who run in built spaces. The secondary hypothesis was that emotional intelligence, psychological well-being, and satisfaction with body image are predictors for running in natural spaces.

## 2. Materials and Methods

A quantitative, cross-sectional study was carried out.

### 2.1. Participants

The sample size was estimated using a 95% confidence, statistical power of 80%, and a correlation coefficient of 0.21. The sample size included 139 male participants and 139 female participants. Excluded were underage runners, those who were not currently running, and those who ran at a similar frequency in natural and constructed areas. A non-probabilistic sample of 331 runners from 20 Mexican states (55.3% women), between 18 and 80 years old (m = 37.4; SD = 11.5), was included. Participants had a mean of 7 years’ running experience (SD = 9.3) and ran for a mean of 3.4 days per week (SD = 1.5). Of the selected participants, 50.8% were couples, 68.9% had paid jobs, 15.7% were students, 6.6% were housewives, 2.7% were unemployed, 2.7% were pensioners or retired, and 3.3% did not respond.

### 2.2. Instruments

General data and characteristics of the practice of physical activity were collected using a form designed for this purpose, and open-ended questions were offered. Natural spaces were considered to be those with a predominance of vegetation and minimal or no presence of human constructions, while built spaces were considered to be those with a predominance of asphalt and a minimal presence of vegetation. Runners were asked about the type of space in which they most frequently ran, without differentiating natural areas from urban natural areas, although they might have occasionally run in another type of space in the past month.

The Brief Emotional Intelligence Inventory (EQ-i-M20) was applied, containing 20 items on a Likert-type scale with five factors: intrapersonal, interpersonal, stress management, adaptability, and general mood. Each item had four response options: never happens to me, sometimes happens to me, almost always happens to me, and always happens to me. In the stress management subscale, a higher score meant lower intelligence; in the rest of the subscales, the scores were straightforward. This structure showed acceptable reliability coefficients for all of the dimensions (ω > 0.700) [31].

The Body Shape Questionnaire (BSQ), validated in the Mexican population, was used to measure unidimensional dissatisfaction with body image. It consists of 18 items on a 6-point Likert-type scale, in which higher scores reflect greater dissatisfaction with body image. The unidimensional structure had high reliability coefficients (ω = 0.947; α = 0.957) [32].

The Ryff Psychological Well-Being Scale, unidimensional version, validated in Mexican university students, was used. It includes 19 items in Likert-type response format, with scores ranging from 1 to 6; higher scores reflect greater psychological well-being. It had high reliability coefficients (ω = 0.937; α = 0.909) [33].

### 2.3. Procedure

A call was made through Facebook groups of Mexican runners, from 1–30 April 2021 (spring). Of those persons called, 98.2% agreed to participate. Participants initially provided the requested informed consent and subsequently answered the questionnaire, using Google Forms.

### 2.4. Data Analysis

Considering sex differences in sports performance, contact with natural spaces, body image dissatisfaction, psychological well-being, and emotional intelligence, all analyses were done independently for men and women. The normality of the data, within groups, was checked by means of skewness and kurtosis, and the multicollinearity was estimated through variance inflation factor (VIF) values. Comparisons between those who ran in natural spaces and those who ran in built spaces were made using student’s *t*-test and chi-square, while the effect size was estimated in both cases (phi and Cohen’s d, respectively). Pearson correlation coefficients were also estimated to identify associations. Finally, multivariate logistic regression models were developed to detect predictors of running in natural spaces, including the variables that had shown significant differences in the comparisons by type of running space.

## 3. Results

### 3.1. Sample Characteristics

Men reported a higher frequency of running in natural spaces (66.2%) than women did (55.2%) (*p* = 0.027). In the group of men, no differences in running performance were found. In women, those who run in natural spaces were found to have greater participation in running and other sports compared to those who run in built spaces (Table 1).

### 3.2. Comparisons According to Exercise Space

Men who run in natural spaces have greater psychological well-being, satisfaction with body image, and emotional intelligence (intrapersonal and stress management) and run more days and kilometers per week than those who run in built spaces. Women who run in natural spaces have lower emotional intelligence (stress management) and run more minutes per occasion (Table 2).

### 3.3. Associations between Study Variables

In men who run in natural areas, a medium correlation was found between psychological well-being and body dissatisfaction and the number of days they run per week. Likewise, there were small correlations of emotional intelligence with psychological well-being, satisfaction with body image, and some running characteristics (Table 3). In women who run in natural spaces, strong and medium magnitude correlations were found between psychological well-being, emotional intelligence (adaptability, general mood, and stress management), and body dissatisfaction; additionally, medium magnitude correlations were found between body dissatisfaction and adaptability. Likewise, age showed a positive relationship of medium magnitude with emotional intelligence. There were small correlations with some characteristics of running, body dissatisfaction, and adaptability (Table 3).

Using a multivariate model, greater psychological well-being and emotional intelligence (stress management) were identified as predictors of running in natural spaces in men. In women, lower emotional intelligence (stress management), running more minutes per occasion, and practicing other physical exercise were identified as predictors (Table 4).

## 4. Discussion

Among all the sports practices, running is one of the most accessible to the population because it is low-cost, requires little equipment, and can be performed in multiple environments such as sidewalks, stadiums, tracks, and natural spaces. Our results show that men run more frequently in natural areas compared to women, which could initially be attributed to a predilection for nature; however, in other contexts, it has been observed that women make more frequent use of parks and natural areas [24]. The difference found may be due to the public insecurity that leads women to avoid these spaces to avoid being victims of violence [34]. However, some gaps remain, since this study did not inquire about the reasons why participants preferred one type of space or another, or about whether they usually run alone or accompanied (which could be a mitigating factor for insecurity), or about the distance of their residence from natural areas, which may be relevant for their use [3].

The first hypothesis sought to determine whether runners in natural environments have characteristics that differ from runners in built environments, focusing separately on men and women. In men, it was found that those who run in natural spaces present greater psychological well-being, less body dissatisfaction, and greater emotional intelligence (intrapersonal and stress management), and they run more days and kilometers per week. These findings on psychological well-being and body dissatisfaction are similar to those from a previous study of Mexican male and female runners [30]. Although the relationship between emotional intelligence and physical activity was already known [18], this is the first study that shows a difference in emotional intelligence according to the type of space in which runners exercise.

On the other hand, women who run in natural spaces participate more in competitions, practice other physical exercise more frequently, and run more minutes per occasion; however, they also present lower emotional intelligence in the stress management dimension. These findings differ from those observed for male participants in a previous study on Mexican male and female runners [30], in which differences for well-being and body dissatisfaction were also expected to be found. It is especially interesting to observe the lower emotional intelligence in the stress management dimension in female outdoor runners, considering that physical activity is associated with higher emotional intelligence [17]. However, differential impacts of stress on physical exercise have been observed. While habitually active individuals exercise more in the face of stress, those who are in the initiation stages of sport exercise less [35]. Additionally, this study reviewed emotional intelligence for stress management and not the presence of stress as in other studies [2].

To our knowledge, this is the first research that reports a relationship between running in natural spaces and lower emotional intelligence in women. This leads us to hypothesize that stress management in women could not only mediate the consolidation of the practice, but also the inclination to exercise in natural spaces as a compensatory measure.

On the other hand, it is interesting to compare the correlation found between age and years of running, since in the men’s group, the correlation is strong (r = 0.607), and in the women’s group, it is weak (r = 0.215). This suggests that men start running at an earlier age and therefore obtain greater cumulative benefits than women. In addition to starting to run at an older age, women may also be intermittent in the practice. This is complemented by the mean number of years running for men (10.9 years), compared to that found in women (5.7 years). These findings are in line with a previous study on adolescents that reported that males engage in more physical activity for longer periods of time than females [20]. Further studies with limited age groups are suggested to thoroughly explore these relationships and detect variations between young adults, middle-aged, and older adult runners.

The secondary hypothesis was that emotional intelligence, psychological well-being, and body dissatisfaction are predictors of running in natural spaces. Regarding males, there were correlations between psychological well-being, body image satisfaction, days running per week, emotional intelligence, and some characteristics of running. These findings add to the available evidence on the relationships between physical activity and mental health [9,10,11,12] and provide new information on running in natural environments. Psychological well-being and emotional intelligence in the stress management dimension are predictors of running in natural spaces for men.

Regarding females, the predictors of running in natural areas were identified as lower emotional intelligence in the stress management dimension, running more minutes per day, and practicing other physical exercise. These two characteristics of running point to consolidated practice; it seems that as women progress in physical exercise, they have a greater predilection for natural spaces. However, this group needs further studies to clarify the controversial relationships between emotional intelligence in the dimension of stress management and adhesion to running. One of the possible hypotheses is based on the evidence for adults raised in areas with higher economic incomes, greater greenness is negatively related to emotional intelligence [14]. This is not possible to verify, since our study did not investigate the area of residence or economic incomes. If there is a causal link between nature of exposure and emotional intelligence [14], then green exercise might help women to enhance their ability to use and manage emotions.

Considering the differences found between men and women, it is necessary to deepen the understanding of the relationships found here through longitudinal studies that allow the establishment of causal relationships. In addition, the study was carried out in a short timeframe over the spring of 2021, and it is possible that the associations found here will be significantly modified in other environmental conditions less favorable for running. Finally, further studies should consider some sociodemographic variables as potential confounding factors with a larger sample size to be incorporated in multivariate models.

## 5. Conclusions

In this heterogeneous sample, men who run in natural areas have greater well-being, less body dissatisfaction, more emotional intelligence (intrapersonal and stress management), and run more days and kilometers per week. Women who run in natural areas have lower emotional intelligence (stress management), run more minutes per occasion, and more frequently practice other sports and participate in competitions. These differences also occur in the associations; while for men running in natural areas is related to greater emotional intelligence, for women the association is negative. Natural environments seem to be related to better performance and some psychological indicators of runners, but these relationships differ between men and women, so further studies are required to deepen the study, especially in regard to emotional intelligence.

## Figures and Tables

**Table 1 ijerph-19-06019-t001:** Sample characteristics according to running space.

	Natural Spaces		Built Spaces			
	*n*	%	*n*	%	*p*	ES (φ)
Living in a couple						
Men	44	44.9	27	54.0	0.303	−0.086
Women	54	54.4	43	52.4	0.999	0.010
Participation in competitions						
Men	58	59.2	27	54.0	0.600	0.050
Women	70	69.3	43	52.4	0.022	0.173
Running for health reasons						
Men	80	81.6	41	82.0	0.999	−0.004
Women	85	84.2	65	79.3	0.442	0.063
Running for aesthetic reasons						
Men	15	15.3	12	30.0	0.260	−0.106
Women	16	15.8	15	18.3	0.695	−0.032
Runs for competitive reasons						
Men	16	16.3	7	14.0	0.813	0.030
Women	15	14.9	6	7.3	0.161	0.118
Running for recreation						
Men	54	55.1	24	48.0	0.487	0.067
Women	55	54.4	43	52.4	0.882	0.020
Other physical exercise						
Men	66	67.3	31	62.0	0.584	0.053
Women	80	79.2	49	59.8	0.005	0.212

ES = Effect size.

**Table 2 ijerph-19-06019-t002:** Comparison of well-being, satisfaction, emotional intelligence, and sports practice by sex, according to running space.

**Variables**	**Men (*n* = 148)**					
	**Natural Spaces (*n* = 98)**		**Built Spaces (*n* = 50)**			
	**Mean**	**SD**	**Mean**	**SD**	** *p* **	**ES (d)**
Psychological well-being	104.2	12.3	95.3	17.8	0.000	0.582
Body dissatisfaction	32.7	15.5	42.9	20.0	0.001	−0.570
EI Intrapersonal	11.1	3.1	9.8	3.1	0.018	0.419
EI Interpersonal	11.1	2.8	11.4	2.1	0.441	−0.121
EI Stress management	7.3	2.3	8.5	2.7	0.005	0.478
EI Adaptability	11.9	2.9	11.7	2.4	0.693	0.075
IE General mood	13.2	3.0	12.7	2.4	0.385	0.184
Age	39.8	12.9	35.6	14.4	0.073	0.307
Years running	10.9	12.3	7.2	9.9	0.067	0.331
Days per week	4.1	1.4	3.3	1.5	0.002	0.551
Minutes per day	67.6	24.0	61.1	27.9	0.154	0.250
Minimum weekly kms	24.5	17.5	18.2	16.0	0.037	0.376
**Variables**	**Women (*n* = 183)**					
	**(*n* = 101)**		**(*n* = 82)**			
Psychological well-being	100.6	11.5	98.7	13.2	0.300	0.153
Body dissatisfaction	46.5	19.9	44.6	19.4	0.527	0.097
EI Intrapersonal	11.2	2.7	11.0	2.7	0.620	0.074
EI Interpersonal	11.9	2.2	11.9	2.3	0.920	0.000
EI Stress management	8.8	2.5	8.0	2.4	0.029	0.326
EI Adaptability	11.4	2.4	11.7	2.4	0.443	−0.125
IE General mood	12.9	2.6	12.6	2.7	0.434	0.113
Age	36.2	9.9	36.5	9.6	0.877	−0.030
Years running	5.7	6.2	4.6	6.2	0.199	0.177
Days per week	4.0	1.4	3.9	1.5	0.762	0.069
Minutes per day	66.6	23.5	57.4	21.6	0.008	0.408
Minimum weekly kms	17.1	14.9	16.0	13.9	0.599	0.076

EI: Emotional intelligence; ES = Effect size.

**Table 3 ijerph-19-06019-t003:** Association between psychological well-being, body dissatisfaction, emotional intelligence, and characteristics of running in natural areas.

	1	2	3	4	5	6	7	8	9	10	11
1. Psychological well-being											
2. Body dissatisfaction											
Men	0.456 **										
Women	−0.431 **										
3. EI Intrapersonal											
Men	0.090	0.118									
Women	0.280 **	−0.042									
4. EI Interpersonal											
Men	0.014	0.221 *	0.460 **								
Women	0.292 **	0.022	0.411 **								
5. EI Stress management											
Men	−0.204 *	0.206 *	−0.050	−0.018							
Women	−0.325 **	0.263 **	−0.287 **	0.054							
6. EI Adaptability											
Men	0.163	0.151	0.597 **	0.703 **	0.006						
Women	0.502**	−0.309 **	0.602 **	0.462 **	-						
7. EI General mood											
Men	0.132	−0.016	0.595 **	0.599 **	0.047	0.812 **					
Women	0.607 **	−0.284 **	0.498 **	0.357 **	−0.345 **	0.660 **					
8. Age											
Men	−0.019	−0.197	−0.066	−0.170	−0.163	−0.231 *	−0.199 *				
Women	0.151	−0.033	0.375 **	0.168	−0.390 **	0.310 **	0.302 **				
9. Years running											
Men	0.102	−0.188	−0.050	0.019	−0.137	−0.024	−0.027	0.607 **			
Women	0.192	−0.114	0.214 *	0.084	−0.270 **	0.146	0.145	0.215 *			
10. Days per week											
Men	0.350 **	−0.230 *	0.107	−0.014	−0.195	0.121	0.085	0.160	0.227 *		
Women	0.118	0.075	0.178	0.129	−0.079	0.269 **	0.173	0.235 *	0.257 **		
11. Minutes per day											
Men	0.213 *	−0.066	−0.109	−0.066	−0.110	−0.126	−0.072	0.033	0.092	0.106	
Women	0.017	0.178	−0.034	0.063	−0.111	−0.132	−0.031	0.167	0.068	−0.073	
12. Minimum weekly kms											
Men	0.234 *	−0.074	0.028	0.041	−0.137	0.047	0.076	0.083	0.211	0.602 **	0.706 **
Women	0.090	0.202 *	0.090	0.168	−0.105	0.152	0.061	0.290 **	0.191	0.599 **	0.609 **

EI: Emotional intelligence. * *p* < 0.05; ** *p* < 0.01.

**Table 4 ijerph-19-06019-t004:** Multivariate model for the detection of predictor variables for running in natural areas, by gender.

Variables	Crude Model			Adjusted Model		
	OR	CI 95%	*p*	OR	CI 95%	*p*
**Men**						
Psychological well-being	1.02	(0.99–1.05)	0.253	1.04	(1.01–1.07)	0.008
Body dissatisfaction	0.98	(0.96–1.01)	0.123			0.100
EI Intrapersonal	1.09	(0.96–1.25)	0.186			0.161
EI Stress management	0.88	(0.76–1.03)	0.103	0.86	(0.74–0.99)	0.038
Days per week	1.23	(0.89–1.69)	0.205			0.072
Minimum weekly kms	1.00	(0.97–1.03)	0.957			0.293
					**R^2^ = 0.207**	
**Women**						
EI Stress management	1.17	(1.02–1.35)	0.030	1.16	(1.01–1.34)	0.036
Minutes per day	1.02	(1.00–1.03)	0.027	1.02	(1.01–1.04)	0.014
Participation in competitions	1.92	(0.98–3.75)	0.058			
Other physical exercise	2.93	(1.43–6.01)	0.003	3.04	(1.49–6.22)	0.02
					**R^2^ = 0.189**	

EI: Emotional intelligence; OR: Odss ratio; CI = Confidence interval.

## Data Availability

The database is available in base abierta.sav.

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
