# Peer review of "Running in Natural Spaces: Gender Analysis of Its Relationship with Emotional Intelligence, Psychological Well-Being, and Physical Activity"

_ijerph, 2022, doi:10.3390/ijerph19106019_

Round 1

Reviewer 1 Report

This study may contribute to the area of knowledge regarding green exercise, but some issues should be noticed. I offer the following comments for the authors’ consideration.

Introduction

The authors have collected some clues on gender differences in psychological indices. I think more information on green exercise will be helpful. For example, some studies have suggested that gender/sex differences may affect green space utilization.

Methods

I suggest the authors describe their experimental settings in detail. When did the authors carry out the investigation? How long did it last? “Natural spaces” usually change with time/season, and their impact on people may also change, so please generally describe the greenness of the “natural spaces” during the investigation.  

Just online recruitment via Facebook without a design for the sampling? Did the authors have an inclusion/exclusion criterion?

Usually, runners may visit both natural and built environments in their running experiences. How did they report their exercise places for this research? For example, a runner may select natural spaces today and built environments on the other day, or a single bout of running exercise can involve both natural and built environments.

What was the timeframe for running in natural or built spaces? In the past week or month?

Did the authors investigate the frequency and duration of running exercise? How did they offer questions, and how did they collect data (e.g., use Likert-scale or open response)?

The authors have defined “Natural spaces” as “those with a predominance of vegetation and minimal or no presence of human constructions.” What about urban green spaces? How could their participants indicate if the exercise was in urban green spaces or natural environments outside the city?

Have the authors considered any control variables for analysis/regression? Such as participants’ sociodemographic information.

Please report the internal consistencies of the used instruments.

Analysis

Did the authors test normality before performing the Pearson correlation and Student's t-test? A multicollinearity test may be necessary before the multivariate logistic regression.

Please indicate the overall interpretation accuracy of the model in the results.

The authors removed some variables from crude models based on their p-values. “Blank values appear in the variables that were discarded from the equation because they were not significant.” These are not very clear to me, so I suggest further illustrating the procedure in the Data Analysis part and citing some relevant studies.

Discussion

What can exact strategies be proposed for the promotion of running based on the current findings?

Reviewer 2 Report

Dear authors, 

The paper entitled "Running in Natural Spaces: Gender Analysis of its Relationship with Emotional Intelligence, Psychological Well-being, and Physical Activity  has been revised. 

In general, I would like to congratulate you on your work. I think the introductory section is very appropriate. However, in my opinion, there are important shortcomings in the method, which I do not know if the authors will be able to remedy. 

My main concern is with the sample used. I have observed a high heterogeneity in the sample that could condition the results obtained. In addition, I do not see data related to statistical power.

Is it possible to compare the results obtained by a young person of 18 with an adult of 80? I have serious doubts

It would be interesting to add the validity and reliability values of the instruments used. I do not see them in any of them.

The method by which the sample was contacted has shortcomings. How many people were contacted? How many answered the questionnaire? For how many days? How can we confirm that there is no bias? 

Then in your sample you describe a high variety of subjects, retired, students, unemployed, etc. However, the analysis only responds to the gender variant.

The discussion section is rather poor, I think it should be expanded.

The same is true for the conclusion. In fact, it seems to me that they make very general statements for the small and heterogeneous sample they present. For example, they state that Consequently, strategies for the promotion of running should be adapted by gender. This idea is not discussed in depth in the discussion.

For all these reasons, I consider that the work is still far from being publishable in its current format. I have considered the option of a major revision.

I hope that my comments will help you to improve your manuscript.

Round 2

Reviewer 1 Report

Thanks for the authors' efforts to improve the manuscript. Since this study only involved a small sample size and there is heterogeneity in the participants, I suggest mentioning their limitations in the Abstract and implying to what extent they can generalize the findings to the general public.

Reviewer 2 Report

The authors have responded to my request, however some improvements must be implemented in order to publish the paper. 

The introduction should be improved as it is a bit poor. I think it could be interesting to make a section on how physical exercise and Physical Education can contribute to the sustainable development of the planet. Review recent literature This new approach is being raised in recent years and could help to make the contribution of the paper even more valuable. 

Please calculate the effect size for the tables presenting the p-value.

The discussion needs to be improved and expanded. Please review better the previous literature and try to rely more on it to explain the results obtained.

My proposal is minor revision of the paper. 
